# Multi-Omic Analysis of the Differences in Growth and Metabolic Mechanisms Between Chinese Domestic Cattle and Simmental Crossbred Cattle

**DOI:** 10.3390/ijms26041547

**Published:** 2025-02-12

**Authors:** Jie Wang, Jiale Ni, Xianbo Jia, Wenqiang Sun, Songjia Lai

**Affiliations:** College of Animal Science and Technology, Sichuan Agricultural University, Chengdu 611130, China; wjie68@163.com (J.W.); njl2077@163.com (J.N.); jaxb369@sicau.edu.cn (X.J.); wqsun2021@163.com (W.S.)

**Keywords:** Chinese domestic cattle, Simmental crossbred cattle, short-chain fatty acids (SCFAs), 16sRNA sequencing, transcriptome, metabolome

## Abstract

In livestock production, deeply understanding the molecular mechanisms of growth and metabolic differences in different breeds of cattle is of great significance for optimizing breeding strategies, improving meat quality, and promoting sustainable development. This study aims to comprehensively reveal the molecular-level differences between Chinese domestic cattle and Simmental crossbred cattle through multi-omics analysis, and further provide a theoretical basis for the efficient development of the beef cattle industry. The domestic cattle in China are a unique genetic breed resource. They have characteristics like small size, strong adaptability, and distinctive meat quality. There are significant differences in the growth rate and meat production between these domestic cattle and Simmental hybrid cattle. However, the specific molecular-level differences between them are still unclear. This study conducted a comprehensive comparison between the domestic cattle in China and Simmental crossbred cattle, focusing on microbiology, short-chain fatty acids, blood metabolome, and transcriptome. The results revealed notable differences in the microbial Simpson index between the domestic and Simmental crossbred cattle. The differential strain *Akkermansia* was found to be highly negatively correlated with the differential short-chain fatty acid isocaproic acid, suggesting that *Akkermansia* may play a key role in the differences observed in isocaproic acid levels or phenotypes. Furthermore, the transcriptional metabolomics analysis indicated that the differentially expressed genes and metabolites were co-enriched in pathways related to insulin secretion, thyroid hormone synthesis, bile secretion, aldosterone synthesis and secretion, and Cyclic Adenosine Monophosphate (cAMP) signaling pathways. Key genes such as *ADCY8* and 1-oleoyl-sn-glycero-3-phosphocholine emerged as crucial regulators of growth and metabolism in beef cattle.

## 1. Introduction

The growth and metabolism of beef cattle are influenced by a variety of factors, including microorganisms, nutrient metabolism, and genetic variation [1]. Microorganisms can enhance the performance and health status of ruminants [2]. Reasonable nutritional formulation and feeding management can influence the body’s metabolism, leading to a significant improvement in growth rate and feed utilization efficiency [3]. Genetic variation is essential for determining growth and metabolic potential, making the selection of superior varieties and individuals a critical factor [4,5].

As a distinct and valuable genetic breed resource, the domestic cattle in Leibo County, Sichuan Province, China, possess the characteristics of diminutive size, robust adaptability, and exceptional meat quality, thereby holding significant research value and latent potential for further exploration. Simmental cattle attract attention because of their rapid growth, superior meat quality, and high meat production [6,7]. In China, they are often used as a parental breed for high-quality beef cattle. Crossbreeding with domestic cattle breeds helps preserve the high-quality meat genes of domestic cattle. It also improves slaughter performance and increases the yield of high-grade beef. The resulting hybrids have shown remarkable advantages and effectively increased the economic benefits of domestic cattle breeding [8,9].

In comparison with domestic cattle, Simmental crossbred cattle display substantial differences in production performance. The underlying molecular mechanisms responsible for these differences warrant further investigation and elucidation. In this study, short-chain fatty acids and microbial sequencing technology, in conjunction with transcriptome and metabolome sequencing technologies, were utilized to explore the disparities in fecal microbial community structure, transcriptional expression, and metabolic pathways between local cattle and Simmental crossbred cattle in Leibo County. This enabled an in-depth examination of the uniqueness of native cattle among domestic animals, the revelation of their distinctive gene regulatory loci and metabolic mechanisms, and the clarification of their specific molecular-level differences from Simmental crossbred cattle.

## 2. Results

### 2.1. Microbial Composition

At the phylum level, the primary observed microbial phyla included *Firmicutes*, *Bacteroidota*, *Euryarchaeota*, *Spirochaetota*, and *Verrucomicrobiota*. At the genus level, *Bacteroides*, *Alistipes*, *Monoglobus*, *Methanobrevibacter*, and *Romboutsia* were the most prevalent. Notably, *Romboutsia* is relatively well represented in the T8 sample (Figure 1a,b). Short-chain fatty acid results showed that AA, BA, and PA levels were relatively high in both TN and LB groups. But there was a significant difference in 4—MVA (*p* < 0.05), with higher levels in the TN group (Figure 1c; see Appendix A for details). ASV-based PCA analysis showed that the overall distribution of fitting circles between the LB group and the TN group was located on both sides of the longitudinal axis. The sample distribution in the LB group was more concentrated, while that in the TN group was relatively dispersed (Figure 1d). A highly significant difference in the Simpson index was observed by the rank-sum test (*p* < 0.01). Each diversity index was listed in Appendix A. After conducting LEfSe analysis (with an LDA threshold of 4.0), a comparison of the relative abundance of bacteria in the LB and TN groups revealed significant differences in the following microbiota: In the TN group, the flora with significant differences included *Clostridia* at the class level, *unidentified Clostridia* at the order level, *Oscillospiraceae* at the family level, *unidentified Oscillospiraceae* at the genus level, and *gut metagenome* at the species level. In the LB group, the flora with significant differences included *Treponema*, *Akkermansia*, and *unidentified Clostridiaceae* at the genus level, metagenome at the species level, Spirochaetaceae at the family level, Spirochaetales at the order level, and several specific species and family classifications. *Clostridiales bacterium enrichment culture clone 06 1235251 76* and *Akkermansiaceae* show statistically significant differences (Figure 1e). The results of differential metabolic pathways calculated by the metagenomeSeq method showed that the TN group was significantly more effective in tetracycline biosynthesis. The biosynthesis of type II polyketone backbones, biosynthesis of various secondary metabolites Part 1, cell adhesion molecules, and plant hormone signaling pathways are down-regulated relative to LB (Figure 1f).

### 2.2. Transcriptomic Analysis

After conducting a comparative analysis between TN and LB, 1078 differentially expressed genes (DEGs) were successfully identified. Among these genes, 772 were up-regulated, and 306 were down-regulated. The volcano plot in Figure 2a visualizes the distribution of the differentially expressed genes. At the same time, the heatmap in Figure 2b further reveals the expression levels of these genes. At the transcriptome level, the differentially expressed genes (DEGs) identified in the TN group compared to the LB group were significantly enriched (*p* < 0.05) among 525 Gene Ontology (GO) terms (Appendix A), primarily associated with immune receptor cell recombination based on immunoglobulin superfamily domains. The regulation of reactions to biostimuli, lymphocyte-mediated immunity, and leukocyte-mediated immune pathways are illustrated in Figure 2c. Text: DEGs detected in the TN group compared to the LB group were significantly enriched (*p* < 0.05) in 65 KEGG terms (Appendix A). The results of KEGG functional enrichment indicated that the main enrichment was observed in primary immunodeficiency disease, graft-versus-host disease, IgA-producing intestinal immune network, autoimmune thyroid disease, allograft rejection, Staphylococcus aureus infection, asthma, antigen processing and presentation, rheumatoid arthritis, African trypanosomiasis, Chagas disease, malaria, Viral myocarditis, calcium signaling pathway, NF-κB signaling pathway, Yersinia infection, natural killer cell-mediated cytotoxicity, Th1 and Th2 cell differentiation, type I diabetes mellitus, and systemic lupus erythematosus (Figure 2d).

### 2.3. Metabolomic Analysis

After conducting a comparative analysis between TN and LB, a total of 1253 differentially expressed metabolites were successfully identified. Specifically, the expression levels of 288 metabolites showed an upward trend, while 965 showed a downward trend. This result has been visualized by the volcano diagram in Figure 3a. At the same time, the heatmap in Figure 3b further reveals the specific expression levels of these differential metabolites. The KEGG functional enrichment results showed that the differential metabolites were mainly enriched in arachidonic acid metabolism, α-linolenic acid metabolism, glycerophospholipid metabolism, linoleic acid metabolism, retrograde endocannabinoid signaling, steroid hormone biosynthesis, and choline metabolism in cancer (Figure 3c).

### 2.4. Joint Analysis Result

The results showed that isocaproic acid was positively correlated with *Frisingicoccus*, *unidentified rickettsia*, *unidentified Enterobacteriaceae*, *Tannerella* (*p* < 0.05), and *Streptococcus* (*p* < 0.01). It was negatively correlated with *Streptomyces*, *Ruminobacterium*, *yeast*, *treponemal*, and *unidentified Clostridaceae* (*p* < 0.05) and highly significantly negatively associated with *Ackermansia*, *anaerobes*, and *hydro anaerobic bacilli* (*p* < 0.01) (Figure 4). Co-enrichment analysis revealed that the metabolome and transcriptome were enriched in processes such as insulin secretion, thyroid hormone synthesis, bile secretion, aldosterone synthesis and secretion, and the cAMP signaling pathway (*p* < 0.05) (Figure 5).

## 3. Discussion

Leibo domestic cattle are important livestock in Leibo County, Sichuan Province. They are known for their strong adaptability and tolerance to rough feeding. They are small in size, have a variety of coat colors, delicate flesh, and are adept at navigating complex terrain and utilizing domestic forage resources. Simmental crossbred cattle are the offspring of Simmental cattle crossed with native cattle. They combine the advantages of fast growth, large size, and good meat quality, exhibiting a tall, muscular body and high economic value [10]. Simmental crossbreeds usually sell for more, but Lebo’s cattle also meet the local demand for beef. Both have their own characteristics and play an important role in agricultural production and market demand. In order to conserve and utilize these resources, it is necessary to strengthen research and breeding for the sustainable development of the livestock sector.

In this study, the microorganisms in the feces of the TN and LB groups were meticulously identified, complemented by the analysis of short-chain fatty acids, and a comprehensive evaluation was conducted. The results showed a significant statistical difference in the Simpson index within the alpha diversity index (*p* < 0.01). This suggested that various environmental conditions, lifestyle habits, or disease states might influence the gut microbiome. For short-chain fatty acids, only one, isocaproic acid, was detected, showing significant differences between the control and experimental groups. In the growth and metabolism of beef cattle, isocaproic acid, as one of the short-chain fatty acids, might play a role through a variety of mechanisms. Some studies have noted that short-chain fatty acids influence growth metabolism by impacting energy metabolism, fat synthesis, and intestinal homeostasis in animals [11]. However, current research on the effects of this substance on growth metabolism is relatively limited, so it is not clear whether it is a key factor contributing to phenotypic differences. In the LEfSe analysis, a variety of differential strains were identified. Among them, bacteria of the genus *Akkermansia* are thought to be closely related to gut health and metabolism. In mice-related studies, the absence of *Akkermansia* has been found to be associated with intestinal mucosal barrier damage and metabolic inflammation [12]. The deficiency or reduction in this bacterium is associated with a variety of diseases, such as obesity, diabetes, hepatic steatosis, inflammation, and response to cancer immunotherapy. *Akkermansia* plays an important role in maintaining gut health and is currently being developed as one of the next generation of probiotics [13]. Using the metagenomeSeq method, differential metabolic pathways were analyzed. The results suggest that tetracycline biosynthesis might be related to the metabolism and growth of bacteria. The presence of bacteria in the gut might have an impact on the health and growth of beef cattle. The biosynthesis of type II polyketone backbones might be related to specific cellular structures or metabolic pathways that could impact cellular function and metabolism in beef cattle. Subsequently, a joint analysis was conducted to create a cluster heatmap of isocaproic acid and associated microorganisms. Observations have shown the presence of individual microorganisms that are highly associated with isocaproic acid. Among them, isocaproic acid was significantly positively correlated with *Streptococcus* (*p* < 0.01), but negatively correlated with *Akkermansia*, *Anaerovorax*, and *Hydrogenoanaerobacterium* (*p* < 0.01). It is worth noting that *Akkermansia* is not only a distinct strain among these strains but also highly correlated with varying short-chain fatty acids in the joint analysis. This suggested that it might be a crucial strain contributing to isocaproic acid differences or phenotypic variations. However, its specific effects still require further exploration.

In transcriptome analysis, 1078 differentially expressed genes (DEGs) were identified. Among these genes, 772 were up-regulated and 306 were down-regulated. The Gene Ontology (GO) analysis revealed that the identified differentially expressed genes were associated with serine-type endopeptidase activity, serine-type peptidase activity, and serine hydrolase activity. Serine is an amino acid that plays an important role in protein synthesis [14]. The metabolism of serine plays a crucial role in muscle growth and metabolism, influencing various processes like protein synthesis, the regulation of growth hormones, and metabolic pathways. Maintaining proper serine metabolism can help promote healthy muscle growth [15]. GO analysis revealed that, apart from the enrichment of differentially expressed genes in certain immune pathways, the primary metabolic pathways associated with muscle growth and metabolism were those related to serine metabolism and its enzymatic activities. Serine is a metabolite with expanding metabolic and non-metabolic signaling properties that affect macromolecular biosynthesis and functional modifications. It largely influences cell survival and function, which might be key to phenotypic differences between the two [16]. Insulin metabolism has been shown to be highly correlated with serine metabolism, and alterations in serine metabolism might lead to changes in insulin sensitivity [17]. KEGG pathway enrichment analysis revealed that differentially expressed genes (DEGs) were significantly enriched (*p* < 0.05) in 65 KEGG terms, including autoimmune thyroid disease, Th1 and Th2 cell differentiation, and type I diabetes. Enrichment in the immune pathway might be a more adaptable aspect of TN. KEGG enrichment analysis revealed that the two groups did not exhibit significant enrichment in the key pathways influencing growth metabolism, such as the FoxO and PPAR pathways [18]. The variances between the two groups could potentially be attributed to other factors.

In metabolomics analysis, 1253 differential metabolites were successfully identified. Specifically, the expression levels of 288 metabolites showed an upward trend, while 965 showed a downward trend. The results of KEGG functional enrichment analysis revealed that the differential metabolites were primarily enriched in pathways such as arachidonic acid metabolism, α-linolenic acid metabolism, glycerophospholipid metabolism, linoleic acid metabolism, retrograde endocannabinoid signal transduction, steroid hormone biosynthesis, and choline metabolism in cancer. Among them, glycerophospholipid metabolism, linoleic acid metabolism, and steroid hormone biosynthesis have been shown to affect the growth and metabolism of animals. For example, glycerophospholipids are involved in cell signaling and cell survival. The metabolism of beef cattle might impact the structure and function of their cells, consequently influencing growth metabolism. Linoleic acid is involved in cell membrane structure and function [19]. In fish studies, it has been found that the linoleic acid content affects the signaling pathways associated with fatty acid metabolism and glucose metabolism. These changes might subsequently lead to alterations in muscle and protein content [20]. In studies on bovine muscle growth, it has been found that anabolic steroid-promoted bovine muscle growth involves complex interactions of many pathways and receptors. Its biosynthesis might affect growth, metabolism, and immune function in beef cattle [21].

Combined transcriptome and metabolome analysis has significant advantages as it provides comprehensive information on gene expression and metabolic changes, reveals the association between genes and metabolism, and improves the accuracy and reliability of data analysis [22,23]. In the joint analysis of this study, it was found that the pathways co-enriched by the metabolome and transcriptome were completely different from those identified through transcription and metabolic analysis alone. Co-enrichment analysis demonstrated remarkable enrichment in growth-associated pathways, prominently including amino acid metabolism and hormone synthesis. Notably, multiple pathways such as insulin secretion, thyroid hormone synthesis, bile secretion, aldosterone synthesis and secretion, and the cAMP signaling pathway exhibited significant enrichment. The genes involved in the enrichment of each pathway are presented in Appendix A. Taking insulin secretion as an example, insulin is a key factor in regulating cell proliferation, differentiation, apoptosis, glucose transport, and energy metabolism, and plays a crucial role in cell fate determination [24,25]. In this pathway, four genes were identified (SNAP25, KCNMA1, CHRM3, KCNMB3), while three genes were down-regulated (KCNMB4, PLCB1D, *ADCY8*). Subsequently, integrative analysis of genes enriched in other pathways revealed that *ADCY8* was enriched in each of these pathways. It has been confirmed that bile secretion affects fat deposition and meat quality in heifers for each of the other pathways [26]. Thyroid hormones play various roles in fat metabolism, increasing basal metabolism to promote energy expenditure and promoting the oxidation of fatty acids instead of being stored as fats [27]. This might play a role in enhancing intramuscular fat (IMF) deposition in cattle. However, the corresponding molecular mechanisms have not been discovered, and participants in the thyroid hormone signaling pathway are significantly associated with IMF in cattle [28]. As an example, the mRNA expression of the thyroid hormone-responsive protein (THRSP) gene is regulated by thyroid hormones, predominantly in adipose tissue [29], and is generally higher in muscles with higher intramuscular fat (IMF) content than in muscles with lower IMF content [30]. All of these data suggested a role for thyroid hormones in regulating intramuscular fat (IMF) deposition in cattle. The synthesis and secretion of aldosterone, a mineralocorticoid, are responsible for maintaining fluid volume, electrolyte balance, and blood pressure homeostasis [31]. Aldosterone is the primary mineralocorticoid synthesized and secreted in the glomerular zone of the adrenal cortex, primarily in response to angiotensin II, elevated serum potassium levels, and the adrenocorticotropic hormone. It plays a central role in electrolyte and fluid volume regulation and the maintenance of blood pressure homeostasis. It is tightly regulated by the renin–angiotensin–aldosterone system, which is also considered one of the causal relationships between obesity and hypertension [32,33,34]. The regulation of cyclic adenosine monophosphate (cAMP) signaling involves multiple energy-balanced signaling systems that affect metabolism. Additionally, cAMP influences glycogenolysis in skeletal muscle [35]. *ADCY8* was remarkably enriched in multiple pathways mentioned above. This enrichment strongly indicates that *ADCY8* can be regarded as a core gene in regulating the growth and metabolism of beef cattle.

Through screening the co-enrichment pathway and comparing data (selecting genes and metabolites with absolute correlation values greater than 0.8 and statistically significant), it was discovered that *ACDY8* was strongly correlated with 1-oleoyl-sn-glycero-3-phosphocholine. This correlation suggested that these could be regarded as the central gene and metabolite to influence the growth and metabolism of beef cattle. Both the gene and metabolites were significantly down-regulated (*p* < 0.01). The anticipated outcome of this study was to observe significant differences in growth metabolism-related genes and metabolites between the two groups. However, the transcriptome analysis revealed that the differences were primarily focused on immunity. While the metabolome analysis yielded expected results to some extent, no key metabolites were identified. The *ACDY8* gene identified through joint analysis is enriched and down-regulated in several important pathways. This gene might play a crucial role in regulating growth and has been identified in meat sheep in recent studies. It has been found to be involved in the metabolic function of tissue and organ structures, as well as in the secretion of hormones such as aldosterone, cortisol, oxytocin, and the adrenocorticotropic hormone in the endocrine system [36].

1-Oleoyl-sn-glycero-3-phosphocholine is a phospholipid that primarily influences the structural function of cell membranes, fat metabolism, energy metabolism, and choline synthesis [37]. At present, there are few studies on this substance, and it is impossible to confirm its specific effect on the growth of beef cattle. The *ADCY8* gene, along with 1-oleoyl-sn-glycero-3-phosphocholine, might play a crucial role in beef cattle breeding.

## 4. Materials and Methods

### 4.1. Ethics Statement

The authors confirm that this study was performed in accordance with the Guidelines of Good Experimental Practices adopted by the Institute of Animal Science of the Sichuan Agricultural University, Chengdu, China. All experimental protocols involving animals were approved by the Animal Care and Use Committee for Biological Studies, Sichuan Province, China (DKY-B2019302083).

### 4.2. Animal Sample Collection

All samples were obtained from Junhao Co., Ltd. in Leibo County, Liangshan City, Sichuan Province, China. This included 10 Simmental crossbred cattle, designated as from within the farm, and 10 samples of domestic cattle, referred to as collected from local farmers. All cattle were gathered at the farm one month in advance. They are fed TMR feed twice a day, with the diet level set according to the 12-month-old feeding standard. All cattle were female, healthy, and 12 ± 2 months old.

### 4.3. Short-Chain Fatty Acids (SCFAs) in Feces

After thawing the fecal sample, 20 mg was weighed into a 2 mL centrifuge tube. Then, 1000 μL of a 0.5% (*v*/*v*) phosphoric acid solution was added, and the mixture was ground using a ball mill at 30 Hz for 1 min. The sample was vortexed at 2500 r/min for 10 min, after which it was sonicated at 4 °C for 5 min. Next, it was centrifuged at 4 °C and 12,000 r/min for 10 min. After centrifugation, 100 μL of the supernatant was carefully pipetted into a 1.5 mL centrifuge tube. Subsequently, 500 μL of MTBE (CNW Technologies, Berlin, Germany) extractant containing an internal standard was added to the tube with the supernatant. The mixture was then vortexed at 2500 r/min for 3 min, followed by sonication at 4 °C for another 5 min. After that, it was centrifuged again at 4 °C and 12,000 r/min for 10 min. Finally, 200 μL of the supernatant was pipetted into the liner of a vial and stored in a −20 °C freezer until GC—MS/MS analysis.

### 4.4. Microbiome

The total genomic DNA from the samples was extracted using the Cetyltrimethylammonium Bromide (CTAB) method, and its concentration and purity were monitored by a 1% agarose gel. After diluting the DNA to 1 ng/μL, the 16S rRNA/18S rRNA/ITS gene was amplified using specific primers and barcodes. The PCR reaction was performed using Phusion’s^®^ High-Fidelity PCR Master Mix (New England Biolabs, London, UK), forward and reverse primers, and template DNA. The PCR products were detected by electrophoresis on a 2% agarose gel and purified. Sequencing libraries were generated using the TruSeq^®^ DNA PCR-Free Sample Preparation Kit (Illumina, San Diego, CA, USA) and indexed barcodes. Sequencing was performed on an Illumina NovaSeq platform to generate 250 bp paired-end reads. FASTP (v0.22.0) was used for quality filtering, and FLASH (v1.2.11) was used to merge paired-end reads [38]. The UCHIME algorithm was used to detect and remove the chimera sequence [39,40]. The 97% similarity sequences were divided into identical Operational Taxonomic Units (OTUs) using Uparse (v7.0.1001) software, and representative sequences were selected for each OTU for subsequent taxonomic annotations [41]. For 16S analysis, annotation was conducted using the Mothur algorithm and the Silva database [42], available at http://www.arb-silva.de/ (accessed on 11 October 2024). MAFFT (v7.490) was used to study the metabolic relationships between Operational Taxonomic Units (OTUs) by conducting multi-sequence alignment [43]. Operational taxonomic unit (OTU) abundance data were standardized to prepare for subsequent alpha and beta diversity analyses. These standardized data are fundamental for accurately assessing species diversity and community composition differences, minimizing biases from sampling and sequencing variations. Sample species diversity was assessed using QIIME and R (version 4.1.2) software, which included metrics such as observed species, Chao1, Shannon, Simpson, ACE, and Good’s coverage. Beta diversity, both weighted and unweighted, was calculated using UniFrac in QIIME. Differences among samples were analyzed using Principal Component Analysis (PCA) and Principal Coordinates Analysis (PCoA) in R (version 4.1.2). The Unweighted Pair Group Method with Arithmetic Mean (UPGMA) clustering method was employed for hierarchical clustering to interpret the distance matrix.

### 4.5. Transcriptomics

RNA was extracted using the Trizol method and subsequently identified and quantified using Qubit and Qsep400. Polyadenylated mRNA was enriched using Oligo(dT) magnetic beads (China, MCE) and then cleaved into small fragments, which were reverse transcribed into complementary DNA (cDNA). Strand-specific libraries were constructed, followed by sequencing adapter ligation and PCR amplification. The concentration and fragment size were determined using Qubit and Qsep400. Finally, quantitative PCR (Q-PCR) quantification was performed. Illumina sequencing was conducted on qualified libraries based on the principle of sequencing by synthesis. Data analysis included quality control filtration using fastp [44], alignment to reference genomes with HISAT [45], new gene prediction with StringTie [46], gene alignment calculation with featureCounts [47], and quantification using FPKM. Differential expression analysis was performed using DESeq2 [48], with *p*-values corrected accordingly. Enrichment analysis was also conducted. Variable splice events were analyzed using rMATS [49], variation site analysis was performed with GATK (version 4.1.9.0) [50], and annotation was completed using ANNOVAR [51]. Differentially expressed genes were analyzed based on the STRING database [52]. Gene set enrichment analysis was performed using GSEA [53], and weighted gene co-expression network analysis was conducted with WGCNA (version 1.71, mergeCutHeight = 0.25) [54].

### 4.6. Metabolomics

We removed samples from the −80 °C freezer and thawed them on ice until no ice remained. After vortexing for 10 s to ensure thorough mixing, 50 μL of the sample was transferred to a centrifuge tube. Next, 300 μL of acetonitrile–methanol internal standard was added and vortexed for 3 min, and then centrifuged at 4 °C at 12,000 rpm for 10 min. Afterwards, 200 μL of the supernatant was taken and transferred to another centrifuge tube. It was allowed to stand at −20 °C for 30 min. After 3 min of centrifugation, 180 μL of the supernatant was transferred to the inner liner of the injection bottle for analysis. The column used was a Waters ACQUITY Premier HSS T3 Column with a particle size of 1.8 μm and dimensions of 2.1 mm × 100 mm. The mobile phase A consists of 0.1% formic acid in water, while mobile phase B consists of 0.1% formic acid in acetonitrile. The column temperature was maintained at 40 °C, the flow rate was set to 0.4 mL/min, and the injection volume was 4 μL. Mass spectrometry data were converted to the mzXML format using ProteoWizard [55]. Peak extraction, alignment, and retention time correction were performed using the XCMS program [56]. Peaks with a deletion rate exceeding 50% were filtered out, the K-nearest neighbors (KNNs) method was used to fill in blank values, and support vector regression (SVR) was employed to correct the peak area. Metabolites were identified by searching self-constructed databases, public databases, prediction databases, and using metDNA methods. Substances with a comprehensive score of 0.5 or higher and a coefficient of variation (CV) value of less than 0.3 in quality control (QC) samples were extracted. The results from both positive and negative modes were combined, retaining only the substances with the highest qualitative grade and the lowest CV value.

### 4.7. Joint Analysis

To intuitively reflect the similarities and differences in the expression patterns of different microorganisms and different metabolites, Spearman correlation hierarchical cluster analysis was performed on different microorganisms and different metabolites. The closer the branches are, the more similar the expression patterns of microorganisms or metabolites. Heatmaps were plotted using the ComplexHeatmap package in the R software (version 4.4.2). According to the results of the KEGG enrichment analysis of differential metabolites and differential genes, the KEGG pathway of the two omics was identified, and a bubble map was created using the KEGG pathway that was co-enriched by the two omics.

## 5. Conclusions

In this study, by integrating multi-omics approaches including 16S microbial sequencing, short-chain fatty acid assay, and transcriptome and metabolome analysis, we have successfully established a comprehensive research framework for exploring the growth and metabolic mechanisms in beef cattle. This integrated methodology not only enabled us to systematically analyze the complex interactions between the gut microbiota, metabolites, and gene expressions but also provided a novel perspective on understanding the regulatory network underlying beef cattle metabolism.

## Figures and Tables

**Figure 1 ijms-26-01547-f001:**
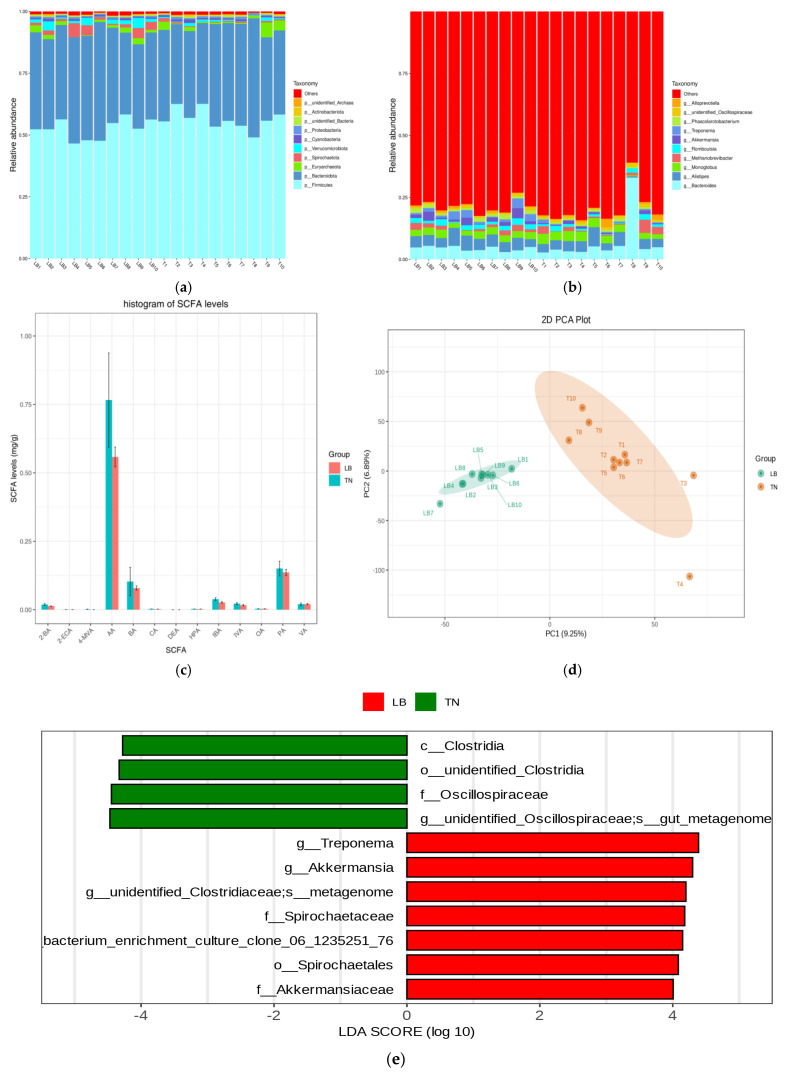
Microbial composition and SCFAs content of TN and LB. The species composition of fecal microorganisms in the TN group and LB group covered two levels: phylum (**a**) and genus (**b**). (**c**) Short-chain fatty acids of TN and LB, 2-BA (2-Methylbutyric acid), 2-ECA (2-Ethylcaproic acid), 4-MVA (Isocaproic acid), AA (Acetic acid), BA (Butyric acid), CA (Caproic acid), DEA (Decanoic acid), HPA (Heptanoic acid), IBA (Isobutyric acid), IVA (Isovaleric acid), OA (Octanoic acid), PA (Propionic acid), VA (Valeric acid). (**d**) Microbial principal component analysis of TN and LB. (**e**) Differential microorganisms with LDA score greater than the set value (default setting of 4). (**f**) ASV-based histogram of PICRUSt2 differential pathway.

**Figure 2 ijms-26-01547-f002:**
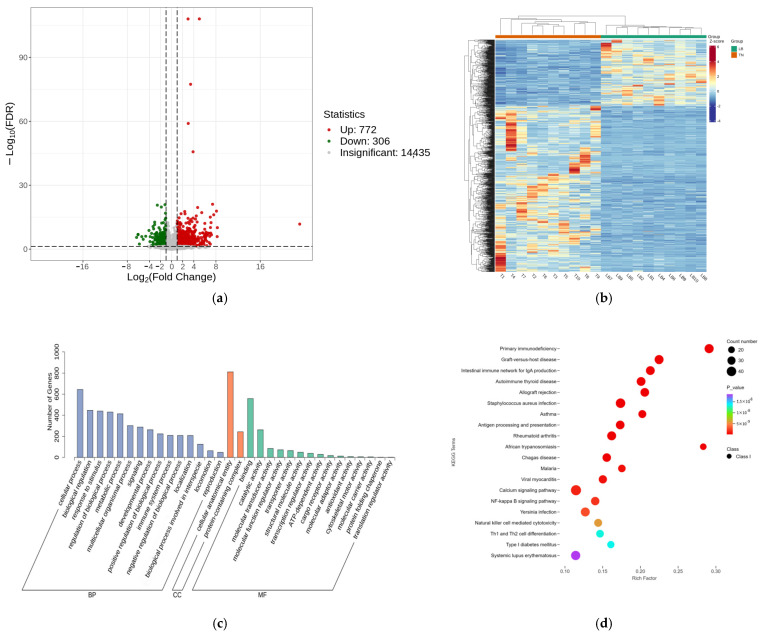
Number of differentially expressed mRNAs in the control/experimental TN group compared to the LB group. (**a**) Volcano diagram of DEGs. (**b**) Hierarchical clustering of DEGs. (**c**) GO item enrichment of DEGs in the TN group compared with the LB group. (**d**) KEGG term enrichment of DEGs in the TN group and LB group.

**Figure 3 ijms-26-01547-f003:**
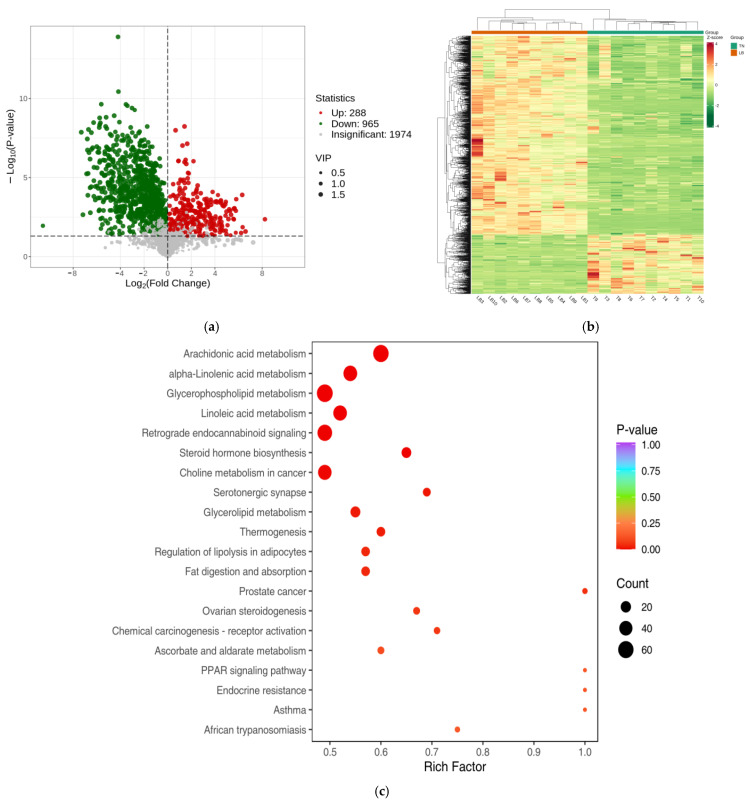
Shows the number of differential metabolites in the control/experimental TN group compared to the LB group. (**a**) Volcano diagram of differential metabolites. (**b**) Hierarchical clustering of differential metabolites. (**c**) KEGG term enrichment analysis of differential metabolites in the TN group and LB group.

**Figure 4 ijms-26-01547-f004:**
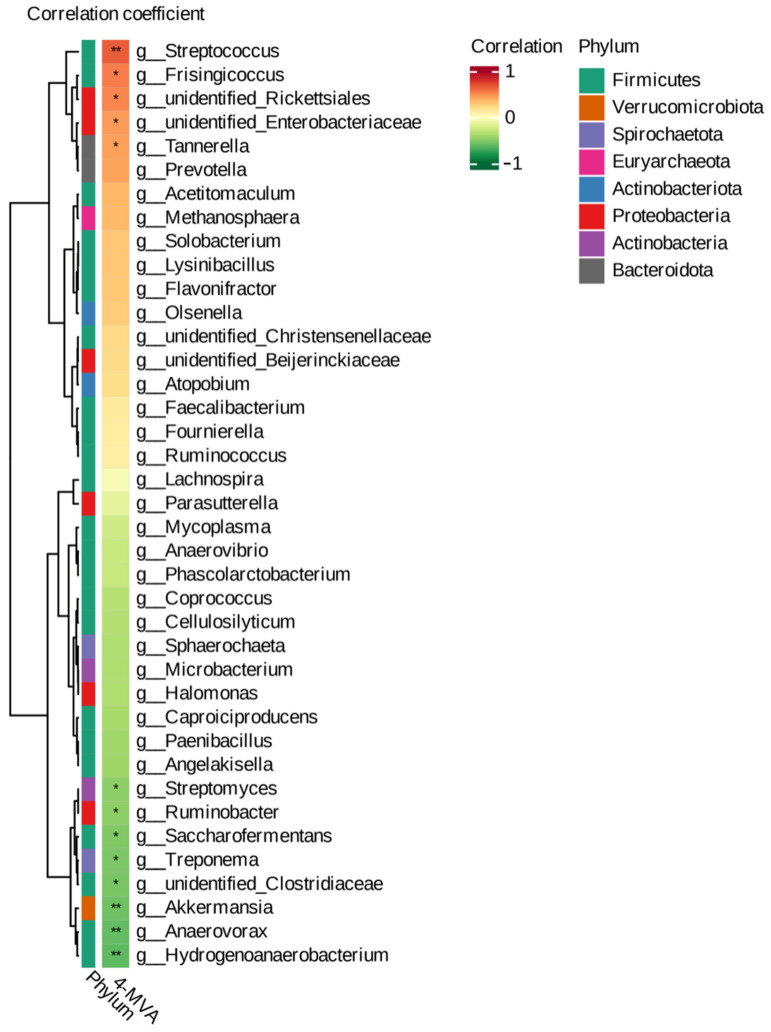
Correlation heatmap between differential SCFAs isocaproic acid concentrations and microbial species composition. Rows represent microorganisms, and columns represent metabolites. The phylogenetic tree on the left represents the hierarchical clustering results of microorganisms, while the phylogenetic tree on the upper side represents the hierarchical clustering results of metabolites. Red indicates a positive correlation, and green indicates a negative correlation. The *p*-value < 0.05 of the significance test of the correlation coefficient indicates a significant difference, denoted by “*”, while the *p*-value < 0.01 indicates a very significant difference, denoted by “**”. The figure shows the genus level.

**Figure 5 ijms-26-01547-f005:**
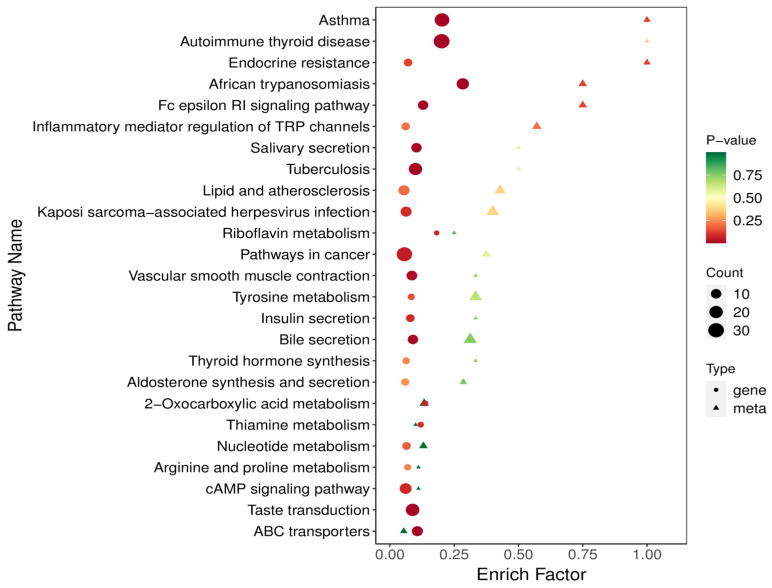
KEGG bubble diagram of the co-enrichment pathway of transcriptome and metabolome. The abscissa represents the enrichment factor (Diff/Background) of the pathway in different omics, and the ordinate represents the KEGG pathway name. The red–yellow–green gradient represents the change in the significance of enrichment from high–medium–low, which is represented by the *p*-value. The shape of the bubble represents different omics, and the size of the bubble represents the number of differential metabolites or genes; the larger the number, the larger the dots.

## Data Availability

All the figures and tables used to support the results of this study are included.

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
