# Peer review of "Multi-Omic Analysis of the Differences in Growth and Metabolic Mechanisms Between Chinese Domestic Cattle and Simmental Crossbred Cattle"

_ijms, 2025, doi:10.3390/ijms26041547_

Round 1
Reviewer 1 Report
Comments and Suggestions for Authors
Microbiome, transcriptome, and metabolomics analysis of dif-2 ferences in growth and metabolic mechanisms between Chi-3 nese domestic cattle and Simmental crossbred cattle
leibo domestic cattle; simmental crossbred cattle; SCFAs; 16sRNA sequencing; tran-34 scriptome; metabolome
There are key words in the title. These should only appear in one of these locations.
The simple summary is unnecessary since it has the Abstract.
The Abstract does not present objectives. The material and method are well written. As well as the results and discussion.
In conclusion, there are results. These should appear in the discussion or methods material. The conclusion must be robust and direct, dealing with facts verified in the research that make it possible to use it, taking into account the title and objective. There should be no results in the conclusion.
No test for plagiarism.
Reviewer 2 Report
Comments and Suggestions for Authors
The authors presented in the current manuscript a microbiome, transcriptome, and metabolomics analysis of differences in growth and metabolic mechanisms between Chinese domestic cattle and Simmental crossbred cattle. Hence providing an understanding of the molecular mechanisms underlying growth and metabolic differences between the cattle. However, I have some feedback that could make this manuscript stronger.
1. The co-enrichment analysis of transcriptome and metabolome pathways is promising but lacks deeper insights into how these pathways interact in vivo to influence phenotypic differences. Authors could integrate the results from transcriptome and metabolome analyses into systems biology models to predict how these pathways influence phenotypic outcomes.
2. The study focuses heavily on molecular and biochemical differences but does not provide a detailed analysis of the resulting phenotypic traits, such as meat quality, growth performance, or feed efficiency, which are critical for practical applications. Authors could relate molecular findings to tangible phenotypic traits like growth rate, feed efficiency, and meat quality to enhance the study's translational value.
3. The results are not cross-validated with independent datasets or cohorts, which would strengthen the reliability of the findings. Authors could perform fecal microbiota transplantation or antibiotic-induced dysbiosis experiments to clarify the causal relationship between gut microbes and phenotypic differences.
Comments on the Quality of English Language
1. Many sentences are overly long and complex, making them harder to follow. Breaking them into shorter, concise sentences would improve readability.
2. Some phrases can be streamlined to avoid redundancy.
3. Some minor grammar issues and awkwardly phrased
